# ADDRESSING EXTRAPOLATION ERROR IN MULTI-AGENT REINFORCEMENT LEARNING

## ABSTRACT

Cooperative Multi-Agent Reinforcement Learning (MARL) has become a critical tool for addressing complex real-world problems. However, scalability remains a significant challenge due to the exponentially growing joint action space. In our analysis, we highlight a critical but often overlooked issue: **extrapolation error**, which arises when unseen state-action pairs are inaccurately assigned unrealistic values, severely affecting performance. We demonstrate that the success of value factorization methods can be largely attributed to their ability to mitigate this error. Building on this insight, we introduce multi-step bootstrapping and ensemble techniques to further reduce extrapolation errors, showing that straightforward modifications can lead to substantial performance improvements. Our findings underscore the importance of recognizing extrapolation error in MARL and highlight the potential of exploring simpler methods to advance the field.

## 1 INTRODUCTION

Cooperative Multi-Agent Reinforcement Learning (MARL) has proven to be a powerful approach for addressing a wide range of complex real-world challenges, including autonomous driving (Zhou et al., 2020), traffic management (Singh et al., 2020), and robot swarm coordination (Hüttenrauch et al., 2017; Zhang et al., 2021a). The complexity inherent in these scenarios poses significant challenges, especially in terms of scalability, as the joint action space expands exponentially with the number of agents. Furthermore, the necessity for decentralized decision-making, grounded in local action-observation histories due to communication constraints, adds to the intricacies of MARL tasks.

While various solutions have been proposed to address these challenges, we identify a critical and often overlooked issue: **extrapolation error** (Fujimoto et al., 2019b). This error arises when unseen or rarely encountered state-action pairs are inaccurately assigned unrealistic values. In the context MARL, where the joint action space is vast, extrapolation error becomes a key limiting factor, especially during training when many actions remain unseen but are required for accurate Temporal Difference (TD) target estimation. For example, in environments like the StarCraft Multi-Agent Challenge (SMAC) (Samvelyan et al., 2019), joint action dimensions can exceed $10^5$, leading to significant errors in estimating the value of these unseen actions.

Although recent MARL approaches, particularly value factorization methods (Son et al., 2019; Sunehag et al., 2017; Rashid et al., 2020b; Wang et al., 2020a), have demonstrated strong performance, much of the focus has been on enhancing the expressive capacity of these models. Early approaches like VDN (Sunehag et al., 2017) and QMIX (Rashid et al., 2020b) were critiqued for their limitations in handling non-monotonic tasks (Mahajan et al., 2019), prompting efforts to improve their representational power. However, despite these advancements, recent studies (Yu et al., 2022; Ellis et al., 2023; Hu et al., 2023) suggest that the performance is not entirely aligned with their expressive capabilities. Notably, QMIX remains the most competitive algorithms across various domains.

In this paper, we propose a shift in focus, emphasizing extrapolation error as a key factor behind the success of value factorization methods. By decomposing the joint Q-function into individual utilities, value factorization significantly reduces extrapolation errors, particularly in environments with large joint action spaces. Additionally, we demonstrate that monotonicity in value factorization plays a crucial role by enabling self-correcting mechanisms in online RL, which help control the propagation of errors. Our analysis reveals that the success of value factorization methods like QMIX is closely tied to their ability to manage extrapolation error.

Building on this insight, we introduce two components: annealed multi-step bootstrapping and ensembled TD targets. These components mitigate bias by incorporating temporally extended trajectories and reduce variance by averaging target values. When applied to existing value factorization methods, these modifications yield substantial performance improvements across tasks, including those in SMAC, Google Research Football (GRF) (Kurach et al., 2020), and SMACv2 (Ellis et al., 2023). Rather than presenting an entirely new algorithm, our goal is to highlight the critical role of extrapolation errors in MARL and showcase how straightforward modifications can yield substantial performance improvements.

## 2 BACKGROUND

### 2.1 DEC-POMDP AND CTDE

We consider Decentralized Partially Observable Markov Decision Process (Dec-POMDP) (Oliehoek & Amato, 2016) in modeling cooperative multi-agent tasks. The Dec-POMDP is characterized by the tuple $\langle \mathcal{N}, \mathcal{S}, \mathcal{A}, r, \mathcal{P}, \mathcal{O}, \mathcal{Z}, \gamma \rangle$, where $\mathcal{N}$ is the set of agents, $\mathcal{S}$ is the set of states, $\mathcal{A}$ is the set of actions, $r$ is the reward function, $\mathcal{P}$ is the transition probability function, $\mathcal{Z}$ is the individual partial observation generated by the observation function $\mathcal{O}$, and $\gamma$ is the discount factor. At each timestep, each agent $i \in \mathcal{N}$ receives a partial observation $o_i \in \mathcal{Z}$ according to $\mathcal{O}(s; i)$ at state $s \in \mathcal{S}$. Then, each agent selects an action $a_i \in \mathcal{A}$ according to its action-observation history $\tau_i \in (\mathcal{Z} \times \mathcal{A})^*$, collectively forming a joint action denoted as $\boldsymbol{a}$. The state $s$ undergoes a transition to the next state $s'$ in accordance with $\mathcal{P}(s'|s, \boldsymbol{a})$, and agents receive a shared reward $r$. The joint action-value function is expressed as $Q^\pi(s_t, a_t) = \mathbb{E}_{s_{t+1:\infty}, a_{t+1:\infty}} \left[ \sum_{i=0}^{\infty} \gamma^i r_{t+i} \right]$, where $\pi$ denotes the joint policy.

This work adheres to the Centralized Training with Decentralized Execution (CTDE) (Oliehoek et al., 2008; Kraemer & Banerjee, 2016) paradigm. In the training phase, CTDE enables policy training to capitalize on globally available information and facilitates the exchange of information among agents. Conversely, during the execution phase, each agent is restricted to accessing solely its individual action-observation history, thereby embodying decentralized execution principles.

### 2.2 VALUE-BASED RL

Value-based RL methods typically involve the iterative adjustment of Q-functions based on the Bellman equation: $Q_{k+1} = \mathcal{T}^\pi Q_k = r + \gamma \mathcal{P}^\pi Q_k$, where $\mathcal{T}^\pi$ denotes the Bellman operator and $\mathcal{P}^\pi Q = \sum_{s'} \mathcal{P}(s'|s, a) \sum_{a'} \pi(a'|s) Q(s, a')$. Restricting the policy to be greedy w.r.t the current Q-function, i.e., $\pi \in \boldsymbol{G}(Q)$, where $\boldsymbol{G}(Q)$ is the set of all greedy policies w.r.t $Q$, transforms the operator into the Bellman optimality operator $\mathcal{T}$, resulting in the Q-learning update $Q_{k+1} = \mathcal{T} Q_k$.

In scenarios with a large state space, the value is often estimated using a differentiable function approximator $Q(s, a; \theta)$ parameterized by $\theta$. Within the framework of deep Q-learning, updates depend on a batch of transitions $(s, \boldsymbol{a}, r, s')$ derived from the replay buffer $\mathcal{D}$. The training of the value function aims to minimize the mean square error:

$$L(\theta) = \mathbb{E}_{(s,a,r,s') \sim \mathcal{D}} \left[ (Q(s, a; \theta) - y)^2 \right], \tag{1}$$

where $y = r + \gamma \max_{a'} Q(s', a'; \theta')$ represents the TD target. The function $Q(\cdot; \theta')$ corresponds to the frozen target network paramertrized by $\theta'$. The periodically updated $\theta'$ ensures a consistent target across multiple iterations.

### 2.3 VALUE FACTORIZATION

Value factorization methods involve learning a factorized value function that encompasses per-agent utilities, denoted as $[Q_i(\tau_i, a_i)]_{i=1}^n$, and is rooted in the principles of Q-learning. A prominently discussed concept in this context is Individual-Global-Max (IGM) (Son et al., 2019), designed to ensure that the locally greedily selected action aligns with the jointly optimal action. Adhering to this constraint, various value factorization methods have been proposed, with some notable examples being VDN (Sunehag et al., 2017), QMIX (Rashid et al., 2020b), QTRAN (Son et al., 2019), and QPLEX (Wang et al., 2020a). In the representation of the joint Q-function, VDN employs an additive assumption: $Q(s, \boldsymbol{a}) = \sum_{i=1}^n Q_i(\tau_i, a_i)$. On the other hand, QMIX utilizes a monotonic mixing

function

$$Q(s, \boldsymbol{a}) = f(s, Q_1(\tau_1, a_1), ..., Q_n(\tau_n, a_n)) \text{ with } \frac{\partial f}{\partial Q_i} \geq 0, \tag{2}$$

where the function $f$ is approximated using a hypernetwork that takes the global state $s$ as input and produces non-negative weights, ensuring monotonicity.

## 3 EXTRAPOLATION ERROR

In practical deep Q-learning with function approximation, the learning of Q-functions may encounter various errors. Following the definition from Anschel et al. (2017), the error $\Delta$ between the current value and the optimal value can be decomposed into three terms:

$$\Delta = Q(s, a; \theta) - Q^*(s, a) = \underbrace{Q(s, a; \theta) - y_{s,a}}_{\text{TAE}} + \underbrace{y_{s,a} - \hat{y}_{s,a}}_{\text{TEE}} + \underbrace{\hat{y}_{s,a} - Q^*(s, a)}_{\text{OD}}, \tag{3}$$

where $y_{s,a} = \mathbb{E}_{\mathcal{D}}\big[r + \gamma \max_{a'} Q(s', a'; \theta)\big]$ is the estimated TD target and $\hat{y}_{s,a} = \mathbb{E}_{\mathcal{D}}\big[r + \gamma \max_{a'} Q(s', a'; \hat{\theta})\big]$ represents the true target with $\hat{\theta} = \arg\min_\theta \mathbb{E}_\pi[(Q(s, a; \theta) - y_{s,a})^2]$.

The first term, **Target Approximation Error (TAE)**, captures the discrepancy between the learned $Q(s, a; \theta)$ and its target $y_{s,a}$. This error can be attributed to factors such as the inexact minimization of $\theta$ through gradient descent and limited representation ability of networks. The second term, **Target Estimation Error (TEE)**, measures the difference between the estimated target $y_{s,a}$ and true target $\hat{y}_{s,a}$, which can be influenced by issues such as overestimation and extrapolation errors. The final term, **Optimality Difference (OD)**, quantifies the gap between the value function of the current policy and that of the optimal policy. Unlike TAE and TEE, which depend on the current Q-function approximation with $\theta$, OD pertains solely to the converged value under the optimal parameter $\hat{\theta}$.

In the context of MARL, especially with value factorization methods, a key distinction from single-agent RL lies in the limited representational capacity introduced by factorization. If the factorization fails to fully capture the current target, TAE arises. As this error accumulates over iterations, it hinders convergence, resulting in a larger OD. To address this, prior research has often focused on improving representation capacity to reduce TAE. Nevertheless, empirical findings from recent works (Yu et al., 2022; Ellis et al., 2023; Hu et al., 2023) suggest that improving representational capacity alone does not always translate into performance gains in complex domains.

We attribute this phenomenon to **extrapolation error**, reflected in the TEE term in (3), which has received little attention in the context of online MARL. Although both TAE and TEE capture Q-function estimation errors, TEE specifically reflects the accuracy of predictions for $Q(s', a')$, while TAE pertains to $Q(s, a)$. This distinction is crucial because $a'$ may not have been observed in the past trajectories, leading to significant extrapolation errors when predicting its value.

In the subsequent sections, we delve deeper into the role of extrapolation error in MARL through theoretical analysis and experimental evidence. By examining error propagation in Q-learning, we demonstrate why ensuring monotonicity is essential for value factorization. Additionally, we revisit several representative MARL algorithms from the lens of extrapolation error, revealing its strong correlation with their performance outcomes.

### 3.1 EXTRAPOLATION ERROR IN MARL

Extrapolation error arises in RL when the value function inadequately estimates the value of actions that are unseen or rare (Fujimoto et al., 2019b). In single-agent setting, if the a state-action pair $(s, a)$ is absent from the dataset, the value $Q(s, a; \theta)$ becomes an uncertain prediction made by the neural network. Consider the update rule $Q(s, a; \theta_{t+1}) \leftarrow r + \max_{a'} Q(s', a'; \theta_t)$, where $Q(s', a'; \theta_t) = Q^\pi(s', a') + e_t(s', a'; \theta_t)$ is decomposed into the true value plus an error term $e_t$. During training, while transition $(s, a, r, s')$ is sampled from the dataset, the next action $a'$ generated by the Q-function/policy may be unseen or rare, potentially leading to a significant $e_t$ for $(s', a')$, propagating a poor estimate to subsequent values.

Extrapolation error is typically associated with offline RL (Fujimoto et al., 2019a; Levine et al., 2020), where the agent operates with a fixed dataset and cannot interact further with the environment, making it likely that the policy samples actions not present in the dataset. In online RL, by contrast,

the policy interacts with the environment and collects data for the actions it generates, allowing for self-correction and reducing extrapolation error. However, recent work (Fujimoto et al., 2023) indicate that as the action space expands significantly, the network becomes increasingly susceptible to erroneous extrapolations, even in online RL. This issue is especially severe in MARL since the action space grows exponentially with agent numbers.

Fortunately, value factorization methods offer an effective solution to this problem. For a factorized Q-function of the form $Q(s, \boldsymbol{a}) = f(s, Q_1(\tau_1, a_1), ..., Q_n(\tau_n, a_n))$, accurate estimation does not require exploration of the entire joint action space. The error $e(s, \boldsymbol{a})$ dependent on joint actions can be decomposed into individual components:

$$Q(s,\boldsymbol{a}) + e(s, \boldsymbol{a}) = f(s, Q_1(\tau_1, a_1) + e_1(\tau_1, a_1), ..., Q_n(\tau_n, a_n) + e_n(\tau_n, a_n)),$$

where the relationship between the errors is approximately given by

$$e(s, \boldsymbol{a}) \approx \sum_{i=1}^{n} \frac{\partial f}{\partial Q_i} e_i(\tau_i, a_i). \tag{4}$$

Using a first-order Taylor expansion, we assume the errors are relatively small compared to the Q-functions. This result implies that maintaining small values for each $e_i(\tau_i, a_i)$ can reduce the overall error in the joint value function. Notably, reducing $e_i(\tau_i, a_i)$ is generally more feasible than directly minimizing $e(s, \boldsymbol{a})$, as $e_i(\tau_i, a_i)$ depends on the number of $a_i$ executed while $e(s, \boldsymbol{a})$ depends on the exact replications of $\boldsymbol{a}$. Consequently, the extrapolation error for a factorized value function can be significantly reduced. Even if certain joint actions are rarely observed, their constituent individual actions may frequently appear in the dataset.

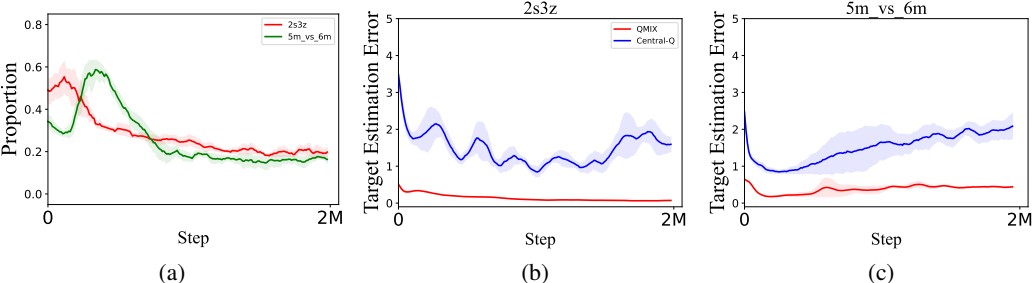

Figure 1: (a) Proportion of unseen state-action pairs on two maps of SMAC. (b)(c) TEE of QMIX and centralized Q-function on two maps of SMAC.

To illustrate this, consider a task with 5 agents, each having 10 possible actions. If the agents select actions uniformly, gathering enough experiences to cover all possible joint actions for a specific state would require over $10^6$ samples in expectation. However, for a factorized value function, only around 30 samples would suffice to ensure reliable estimates that are not merely random guesses. A similar situation arises in SMAC maps such as *2s3z* and *5m_vs_6m*. In Fig. 1(a) we show the proportion of unseen $(s', \boldsymbol{a}')$ in each batch during updates, indicating that 20% to 60% of $\boldsymbol{a}'$ are unseen under $s'$ yet contribute to TD target computation, leading to significant reliance on extrapolated values. Furthermore, by comparing the TEE of QMIX and a centralized Q-function on these maps (Fig. 1(b)(c)), we observe that despite QMIX's limited representational ability, the TEE of the centralized Q-function is substantially larger. These results underscore the importance of managing extrapolation error and highlight the key advantages of value factorization in mitigating this issue. Further details of these examples can be found in Appendix B.1.

## 3.2 PROPAGATION OF EXTRAPOLATION ERRORS

As previously discussed, one critical way to mitigate extrapolation error is through increased execution of unseen actions in the environment, which distinguishes online RL from offline RL. In online RL, extrapolation error is often less severe because once a value is overestimated, the likelihood of selecting the corresponding action increases, making it easier to collect additional samples. These samples help reduce the error by providing more accurate updates. However, in value factorization methods, overestimation of the joint Q-function does not directly lead to an increased probability of choosing the corresponding actions for individual agents. To address this, we introduce an important constraint called Error Propagation Consistency (EPC):

**Definition 1** (EPC). *In value factorization methods, for a joint value function $Q(s, \boldsymbol{a})$, if its corresponding individual utilities $[Q_i(\tau_i, a_i)]_{i=1}^n$ satisfy that, overestimation of $Q(s, \boldsymbol{a})$ will result in the overestimation of all $Q_i(\tau_i, a_i)$, we say that the factorization structure adheres to EPC.*

Value factorization methods require the EPC constraint because the behavioral policy is driven by individual utilities rather than the joint Q-function. Therefore, the factorization structure must propagate errors consistently from the target to the individual utilities, enabling the self-correction mechanism inherent to online RL. In essence, factorization structures that satisfy EPC must be monotonic, as formalized in the following proposition:

**Proposition 1.** *Monotonicity , expressed as $\frac{\partial Q(s, \boldsymbol{a})}{\partial Q_i(\tau_i, a_i)} \geq 0$, stands as a sufficient and necessary condition for EPC.*

To illustrate, consider gradient descent on the objective function $\min \mathbb{E}[(y - f(Q_1, ..., Q_n))^2]$, where $y$ represents the TD target and $Q = f(Q_1, ..., Q_n)$ represents the factorized joint value function. The update rule of individual utility on state-action pair $(s, \boldsymbol{a})$ is given by:

$$Q_i(\tau_i, a_i) \leftarrow Q_i(\tau_i, a_i) + 2\alpha[y - f(s, Q_1(\tau_1, a_1), ..., Q_n(\tau_n, a_n))]\frac{\partial f}{\partial Q_i}|_{s,\boldsymbol{a}},$$

where $\alpha$ is learning rate. For a monotonic $f$ with $\frac{\partial f}{\partial Q_i} \geq 0$, if the target $y$ exceeds the current value function, i.e. $y > f(Q_1, ..., Q_n)$, then the individual utility $Q_i$ will be update to a higher value for $(\tau_i, a_i)$. Conversely, for a non-monotonic function, a larger target $y$ may lead to some smaller $Q_i$.

Now, consider a common scenario where the target Q-value contains an error due to extrapolation, and the max operator causes the target $y$ to be overestimated. For monotonic factorization, each individual utility $Q_i(\tau_i, a_i)$ will be overestimated. Since actions are selected individually based on each agent's utility, this overestimation increases the likelihood of choosing each $a_i$ of $Q_i(\tau_i, a_i)$, thereby making the joint action $\boldsymbol{a} = [a_1, \ldots, a_n]$ under state $s$ more likely. In this scenario, the self-correction mechanism comes into play: as the poorly estimated value of $(s, \boldsymbol{a})$ is revisited more frequently, the corresponding error is updated and reduced. However, if some $Q_i(\tau_i, a_i)$ is underestimated due to $\frac{\partial f}{\partial Q_i} < 0$, the likelihood of selecting the joint action $\boldsymbol{a}$ may decrease, as underestimation of $Q_i(\tau_i, a_i)$ lowers the probability of selecting $a_i$. In such cases, the self-correction mechanism may fail, leading to error accumulation and degraded performance.

In summary, by analyzing the propagation of extrapolation error, we find that monotonicity is essential for preserving the self-correction mechanism in online RL. Similar observations can also be found in Peng et al. (2021) and Hu et al. (2023), where non-monotonic factorization, despite its theoretical correctness and superior expressive abilities, tends to underperform compared to monotonic counterparts, especially in complex tasks.

### 3.3 EXTRAPOLATION ERRORS OF EXISTING APPROACHES

Extrapolation error is seldom addressed in existing online MARL approaches, yet it plays a critical role in shaping performance. Although conducting a unified analysis of existing works through the lens of extrapolation error is challenging—due to differences in learning schemes, structures, and various factors contributing to performance—some general insights can be drawn. Specifically, we observe that approaches that involve learning over the joint action space typically underperform in evaluations. To illustrate this, we provide a brief analysis of existing approaches, and a detailed analysis of QPLEX, which learns joint actions but still achieves relatively good performance.

First, as extensions from single-agent RL, methods like MADDPG (Lowe et al., 2017) and COMA (Foerster et al., 2018) have become popular but tend to underperform in complex tasks. In comparison, methods like MAA2C (Papoudakis et al., 2020) and MAPPO (Yu et al., 2022), which are also derived from single-agent RL, demonstrate a clear performance gap. Moreover, FACMAC (Peng et al., 2021), which applies QMIX's factorization to MADDPG, also exhibits superior performance. A key observation is that both MADDPG and COMA learn joint Q-functions, leading to significant TEE as demonstrated in Fig. 1. By contrast, MAA2C and MAPPO rely on joint value functions, and FACMAC employs factorized Q-functions, which avoid extrapolation over the joint action space. We provide further experimental evidence regarding MADDPG and FACMAC in Section 5.

Second, within value factorization methods, we observe similar issues when joint Q-functions are involved. For instance, QTRAN requires an additional learning step for a centralized Q-function. This extra step undermines the inherent advantage of value factorization, leading to substantial TEE and poor performance in complex tasks, as observed by many prior works (Yang et al., 2020; Wang et al., 2020a; Rashid et al., 2020a). In contrast, the most widely used value factorization algorithm, QMIX, benefits from both factorized action space and monotonicity, despite its limited expressive capacity. As shown in previous studies (Yu et al., 2022; Ellis et al., 2023; Hu et al., 2023), QMIX consistently achieves competitive results across various domains.

Another noteworthy example is QPLEX, which, despite achieving full expressive capability, relies on estimating weights $\lambda_i(s, \boldsymbol{a})$ across the joint action space. From the perspective of extrapolation error, this reliance on $\lambda_i(s, \boldsymbol{a})$ introduces a risk: QPLEX's factorization ensures $\frac{\partial Q}{\partial \lambda_i} \leq 0$, meaning that underestimation of joint Q-function will lead to an overestimation of $\lambda_i$. As discussed in the previous section, underestimating the joint Q-function reduces the likelihood of executing the corresponding joint action $\boldsymbol{a}$, as QPLEX enforces monotonicity by setting $\frac{\partial Q}{\partial Q_i} = 1$. As a result, certain $\boldsymbol{a}$ associated with $\lambda_i(s, \boldsymbol{a})$ will be further extrapolated, accumulating overestimation as training progresses. This issue is evident in Fig. 2, where the solid line (left y-axis) shows the win rate,

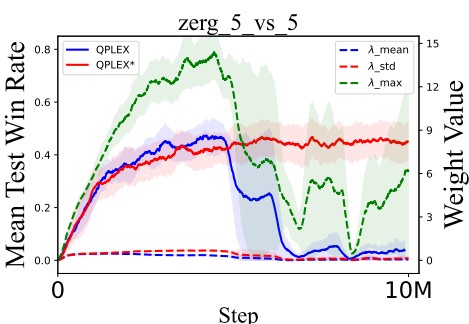

Figure 2: Mean test win rate and the value of $\lambda_i$ of QPLEX on SMACv2.

and the dashed line (right y-axis) represents the value of $\lambda_i$ per batch. The performance of QPLEX degrades midway through training, which we trace to the accumulation of error in $\lambda_i$. While the mean and standard deviation of $\lambda_i$ remain small, the maximum value of $\lambda_i$ grows significantly as training progresses, eventually leading to performance degradation. This implies that QPLEX suffers error accumulation on certain $\lambda_i$ due to extrapolation within the joint action space. To address this issue, we modify QPLEX by bounding $\lambda_i$ using a sigmoid function, resulting in a variant we call QPLEX*. As shown in Fig. 2, this adjustment constrains $\lambda_i$, preventing the instability caused by error accumulation. While this modification limits the expressive power of QPLEX, it does not degrade performance, suggesting that full expressiveness may not be critical in this context. Detailed results are provided in Appendix B.2.

In summary, these findings highlight that extrapolation error is a critical, yet underexplored, issue in MARL. While many existing methods focus on improving performance by addressing TAE, our analysis suggests that mitigating extrapolation error may be more crucial, particularly when efforts to minimize TAE involve learning over the joint action space. Given the significant role of extrapolation error in the performance of MARL methods, a straightforward idea is to prioritize its reduction and evaluate whether this leads to further improvements. In the following sections, we present techniques specifically aimed at reducing extrapolation error and demonstrate their substantial impact on performance.

## 4 ADDRESSING EXTRAPOLATION ERRORS

In this section, we introduce two techniques aimed at reducing the bias and variance of TEE, thereby mitigating extrapolation error. Despite their simplicity, these techniques serves as extensions to existing methods and demonstrate the importance of minimizing extrapolation error. These approaches can, in principle, be integrated with any value factorization, but here we use QMIX as an illustrative example.

### 4.1 ADDRESSING BIAS THROUGH ANNEALED MULTI-STEP BOOTSTRAPPING

Multi-step methods, which incorporate temporally extended trajectories into value updates, have been shown to improve learning efficiency (Sutton & Barto, 2018). In deep RL, when the value network is undertrained, multi-step targets from unbiased samples can mitigate the impact of the value function on the target, thus reducing bias. A representative multi-step method for Q-learning is Peng's $Q(\lambda)$

(PQL) (Peng & Williams, 1994), whose operator $\mathcal{N}_\lambda^{\mu,\pi}$ applicable to any policies $\mu$ and $\pi$, is defined as follows:

$$\mathcal{N}_\lambda^{\mu,\pi} Q = (1-\lambda) \sum_{n=1}^\infty (\lambda \mathcal{T}^\mu)^{(n-1)} \mathcal{T}^\pi Q, \tag{5}$$

where $\lambda \in [0, 1]$. PQL is commonly used in the implementation of many existing MARL algorithms (Peng et al., 2021; Zhang et al., 2021b; Hu et al., 2023), but its properties are seldom discussed and warrant further investigation.

Referring to the error defined in Section 3.1, we consider the following update with error propagation:

$$\pi_k \in \boldsymbol{G}(Q_k) \text{ and } Q_{k+1} := \mathcal{N}_\lambda^{\mu,\pi_k}(Q_k + e_k), \tag{6}$$

where $\mu$ is maintained as a fixed behavior policy (to be discussed later). The following proposition illustrates the error-reducing nature of PQL.

**Proposition 2.** *The target estimation error for each update step $k$ satisfies:*

$$\|Q_{k+1} - \mathcal{N}_\lambda^{\mu,\pi_k} Q_k\|_\infty \leq \beta\epsilon, \tag{7}$$

*where $\epsilon = \|e_k\|_\infty$ and $\beta = \frac{\gamma(1-\lambda)}{1-\gamma\lambda}$. The accumulated error related to $\epsilon$ is $\mathcal{O}(\frac{\gamma(1-\lambda)}{(1-\gamma)^2}\epsilon)$.*

*Proof.* The proof follows from Kozuno et al. (2021), and we provide full details in Appendix C. □

The proposition indicates that extrapolation error and its propagation are linked to $\lambda$. A larger $\lambda$ reduces error, aligning with the intuitive understanding that $\lambda$ controls the balance between collected returns and learned value functions for target estimation.

However, adopting a larger $\lambda$ is not always beneficial. With a fixed behavior policy $\mu$, the PQL operator converges to the fixed point of $\lambda \mathcal{T}^\mu + (1-\lambda)\mathcal{T}$ (Harutyunyan et al., 2016), implying convergence to a biased policy $\lambda\mu + (1-\lambda)\pi$. Although PQL can eventually converge to the optimal policy as $\mu$ approaches $\pi$ (Kozuno et al., 2021), this may not always happen in practice. The behavior policy typically originates from the old policy stored in the replay buffer, which may not align closely with the current policy before convergence. Moreover, behavior policies are often limited to $\varepsilon$-greedy exploration in practical algorithms. Thus, a large $\lambda$ may lead to a highly biased policy and suboptimal performance. Experimental results (Fig. 7 in Appendix) across various $\lambda$ confirm that while a larger $\lambda$ is more efficient during the initial stages of training, its performance may decline, eventually lagging behind smaller $\lambda$ values in later stages.

To address this issue, we propose a $\lambda$ annealing strategy, which leverages the error-reducing properties of PQL with a large initial $\lambda$ while gradually annealing it during training to prevent policy bias. To ensure stability, we update $\lambda_k$ in sync with the fixed target network as follows:

$$\lambda_k = \lambda^* + (1-\lambda^*)/(1+\alpha k), \tag{8}$$

where $\alpha = 10/T$, with $T$ being the total environmental steps for training. This scheme is chosen heuristically, but the algorithm is not very sensitive to how $\lambda$ is annealed, as long as it converges to $\lambda^*$. While $\lambda^*$ would ideally be zero to achieve the optimal policy, practical training conditions may necessitate stopping earlier. For instance, as seen in the *3s5z_vs_3s6z* task (Fig. 5(c)), if the total training steps are insufficient for convergence, maintaining a relatively large $\lambda^*$ proves beneficial.

### 4.2 ADDRESSING VARIANCE THROUGH ENSEMBLED TARGET

Poor value estimation not only introduces bias but also contributes to increased variance. Additionally, similar to GAE (Schulman et al., 2015), PQL reduces bias by introducing variance from Monte Carlo samples. To address this, ensemble learning has become a prevalent strategy in deep learning for reducing variance and improving robustness (Ganaie et al., 2022). By combining predictions from multiple independently trained models, ensemble leverages their diverse perspectives to achieve a more robust and generalized learning process.

Consider $M$ independently estimated Q-functions: $Q(s, \boldsymbol{a}; \boldsymbol{\theta}^j) = Q^*(s, \boldsymbol{a}) + e^j(s, \boldsymbol{a})$, with $e^j$ representing the error term, assumed to be i.i.d across $j$ for each fixed state-action pair. By averaging

these Q-functions, the variance can be reduced proportionally to $1/M$:

$$\text{Var}[\frac{1}{M} \sum_{j=1}^{M} Q(s, \boldsymbol{a}; \theta^j)] = \frac{1}{M} \text{Var}[e^j(s, \boldsymbol{a})].$$

This reduction exploits the i.i.d. nature of the errors $e^j$. Unlike previous works that assume certain additional properties for the error terms (Anschel et al., 2017; Chen et al., 2021), we make no such assumptions, recognizing that the limited expressiveness of value factorization methods may introduce model bias into $e^j$, stemming from TAE as discussed in Section 3. Further details on this are provided in Appendix E.3.

In this approach, we propose directly averaging joint Q-functions. When integrated with QMIX, the ensembled joint Q-function is expressed as:

$$Q(s, \boldsymbol{a}; \bar{\boldsymbol{\theta}}, \psi) = \sum_{j=1}^{M} Q(s, \boldsymbol{a}; \boldsymbol{\theta}^j, \psi) = \sum_{j=1}^{M} H(s, Q_1(\tau_1, a_1; \theta_1^j), ..., Q_n(\tau_n, a_n; \theta_n^j); \psi), \quad (9)$$

where $H$ represents the monotonic hypernetwork parameterized by $\psi$. This ensembled Q-function is then utilized in the PQL target to reduce the estimation variance for rarely seen state-action pairs, as shown in the following loss function:

$$L(\boldsymbol{\theta}, \psi) = \sum_{j=1}^{M} \mathbb{E}_{\mathcal{D}} \left[ (Q(s, \boldsymbol{a}; \boldsymbol{\theta}^j, \psi) - y_{s, \boldsymbol{a}})^2 \right], \quad (10)$$

where $y_{s,\boldsymbol{a}}$ is the PQL target. The theoretical requirements behind these results implicitly assume that the factorization satisfies the EPC condition (i.e., it is monotonic). This is essential because both PQL and ensemble methods control the error of the joint Q-function, and only a monotonic factorization, as seen in Eq. (4), can properly propagate this control to the individual utilities.

We summary the complete algorithm Appendix D, with several important practical considerations. First, the hypernetwork does not take actions as input; thus, using an ensemble of hypernetworks is not expected to help reduce estimation error, as verified in Appendix E.3. Therefore, we use a shared mixing network. Second, in (9), instead of averaging the individual utilities before inputting them into the mixing network, we opt for averaging the joint Q-functions. This deliberate choice is motivated by the belief that the mixing network, when presented with individual utilities exhibiting higher variance, can mimic the effect of target policy smoothing observed in TD3 (Fujimoto et al., 2018). This technique is important in tasks with sparse reward such as GRF, as illustrated in Appendix E.5. Third, double q-learning is employed to the training. It is an important technique to reduce overestimation in Q-learning. While ensemble methods may already help with overestimation (Appendix E.3), we retain it for consistency with previous works.

## 5 EXPERIMENTS

In this section, we evaluate the performance of proposed approach across three domains: SMAC, GRF and SMACv2. Our aim is to demonstrate that, due to the identification of extrapolation error, simple modifications to existing methods can lead to significant performance improvements. Additionally, we conduct ablation studies to elucidate the impact of individual components. Our primary focus is on comparing QMIX, which we refer to as Annealed Ensemble QMIX (AEQMIX). Further results, including those for AEVDN and AEQPLEX*, as well as details regarding hyper-parameters and settings, can be found in Appendix E.

### 5.1 MAIN RESULTS

**StarCraft Multi-Agent Challenge (SMAC)**: The results include evaluations across 15 maps, AE-QMIX generally outperforms QMIX, particularly on challenging maps. Due to the simplicity of some maps, the performance improvement is not significant and, in some cases, as shown in Fig. 8 in Appendix, AEQMIX convergence slower. This is because $\lambda$ annealing will incessantly shift of the fixed point that may slow convergence for simple tasks.

Table 1: Comparison of the mean test win rate between AEQMIX and QMIX on 15 maps of SMAC, 5 maps of GRF and 15 maps of SMACv2.

| Map | AEQMIX | QMIX | Map | AEQMIX | QMIX | Map | AEQMIX | QMIX |
|---|---|---|---|---|---|---|---|---|
| 1c3s5z | **100.0** | **100.0** | 6h_vs_8z | **79.3** | 33.7 | zerg20_vs_23 | **36.6** | 13.9 |
| 2s3z | **100.0** | **100.0** | corridor | **94.6** | 93.0 | protoss5_vs_5 | **76.5** | 69.5 |
| 2s_vs_1sc | **100.0** | **100.0** | MMM2 | **98.9** | 96.5 | protoss10_vs_10 | **83.3** | 70.2 |
| 8m | **100.0** | **100.0** | GRF_3v1 | **78.9** | 46.2 | protoss20_vs_20 | **89.7** | 71.4 |
| MMM | **100.0** | **100.0** | GRF_corner | **22.7** | 19.5 | protoss10_vs_11 | **55.0** | 37.1 |
| 2c_vs_64zg | **99.7** | 98.9 | GRF_ca_easy | **73.2** | 64.5 | protoss20_vs_23 | **42.2** | 16.3 |
| 3s5z | **99.1** | **99.1** | GRF_ca_hard | **49.8** | 44.5 | terran5_vs_5 | **76.9** | 64.4 |
| 3s_vs_5z | **92.8** | 83.9 | GRF_rps | **67.5** | 52.6 | terran10_vs_10 | **79.8** | 66.6 |
| 8m_vs_9m | **91.8** | 90.5 | zerg5_vs_5 | **62.1** | 40.4 | terran20_vs_20 | **69.8** | 54.9 |
| 10m_vs_11m | **96.1** | 94.8 | zerg10_vs_10 | **64.8** | 45.0 | terran10_vs_11 | **66.1** | 40.7 |
| 3s5z_vs_3s6z | **96.9** | 60.2 | zerg20_vs_20 | **55.3** | 33.1 | terran20_vs_23 | **31.4** | 12.5 |
| 5m_vs_6m | **88.6** | 83.2 | zerg10_vs_11 | **49.9** | 26.5 | **Average** | **76.4** | 63.5 |

**Google Research Football (GRF)** (Kurach et al., 2020) poses challenges for value factorization methods due to sparse rewards. Recent work (Papoudakis et al., 2020) highlights the poor performance of value factorization methods in sparse reward scenarios, and QMIX performs poorly in GRF, as tested in (Li et al., 2021; Yu et al., 2022). In this paper, we use a large $\lambda$ in PQL to facilitate the training of value functions. As shown in the table, AEQMIX outperforms QMIX across 5 tasks. It's worth noting that, due to the utilization of a large $\lambda$, the ensembled target may have a limited impact on performance since the majority of the target is derived from the return. Consequently, we opt for an ensemble size of $M = 2$ in GRF. Additional results on GRF are available in Appendix E.5.

**SMACv2** (Ellis et al., 2023) is introduced to address some drawbacks of SMAC, such as the lack of stochasticity and partial observability. In contrast to SMAC and GRF, SMACv2 incorporates randomly generated units and initial positions, introducing more stochasticity and potentially creating scenarios that are exceedingly challenging to win. The algorithms are tested on 15 maps of SMACv2, where QMIX encounters difficulties in achieving a high win rate across these scenarios, particularly in these *20_vs_23* tasks. Notably, AEQMIX exhibits significant improvements across all maps.

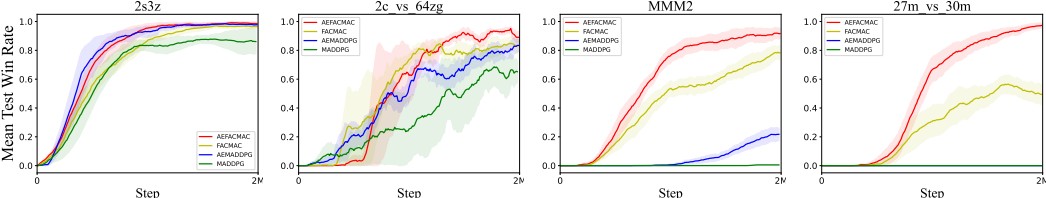

Figure 3: Mean test win rate of AEFACMAC, FACMAC, AEMADDPG and MADDPG on SMAC.

## 5.2 RESULTS FOR POLICY-BASED METHODS

While this paper primarily focuses on value-based methods, it's important to acknowledge that off-policy policy-based methods, such as MADDPG and FACMAC, are also related to extrapolation errors due to their reliance on Q-functions. To illustrate the effectiveness of our approach and the significance of extrapolation error, we present a comparison between MADDPG, FACMAC, and our modified versions, AEMADDPG and AEFACMAC.

The results, as shown in Fig. 3, demonstrate that AEFACMAC outperforms FACMAC, AEMADDPG outperforms MADDPG, and FACMAC outperforms MADDPG. These findings align with our observations regarding extrapolation error; specifically, FACMAC, with its factorized Q-function, exhibits smaller extrapolation error compared to MADDPG, while the AE+ methods show reduced extrapolation error relative to their original counterparts. Additional experiments comparing AEFACMAC and FACMAC can be found in Appendix E.4.

### 5.3 Ablation Studies and Discussions

In this subsection, we conduct further experiments to analyze the impact of the annealed PQL and ensembled target on TEE and performance. The method with a fixed $\lambda$ and ensemble is denoted as EQMIX, while the version without ensemble is referred to as AQMIX. The ensemble size is represented by $M$.

Fig. 4 shows the impact of different $\lambda$ and $M$ on the TEE. First, as expected, a larger $\lambda$ corresponds to lower TEE. Our $\lambda$ annealing approach maintains TEE at a low level throughout training. This helps facilitate more efficient learning during the early stages, while mitigating bias later in training. Second, increasing the ensemble size $M$ reduces variance, consistent with the well-known property of ensemble methods to lower variance by a factor of $1/M$.

Fig. 5(a)(b) illustrate the effects of different sizes and annealing on performance. Firstly, larger ensemble sizes $M$ consistently lead to improved performance, demonstrating the benefits of variance reduction through ensembling. Secondly, comparing AQMIX with QMIX under the same $M$ shows that annealing generally enhances performance. Finally, the combined approach of annealing and ensembling demonstrates a mutually beneficial effect, leading to significant performance improvements. However, it is

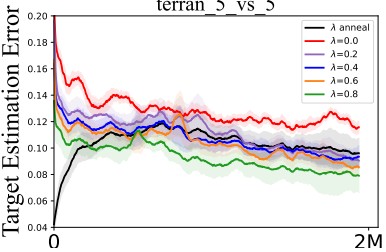

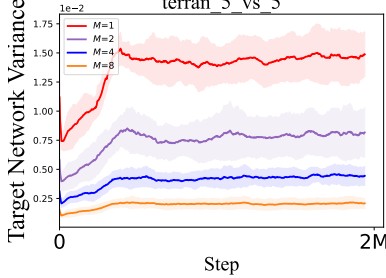

Figure 4: Target estimation error with different $\lambda$ and variance with different ensemble size $M$.

noteworthy that both AQMIX and AEQMIX(M=2) perform worse than QMIX. We attribute this to premature $\lambda$ annealing. As shown in Fig. 5(c), smaller $\lambda$ performs poorly on *3s5z_vs_3s6z* task due to insufficient convergence. Thus, annealing $\lambda$ too early can negatively impact performance. In contrast, when a larger ensemble is used, convergence is achieved earlier on this map, making the annealing of $\lambda$ more appropriate. This in turn benefits training, suggesting that the timing of $\lambda$ annealing should be adjusted based on convergence speed, which is influenced by the ensemble size.

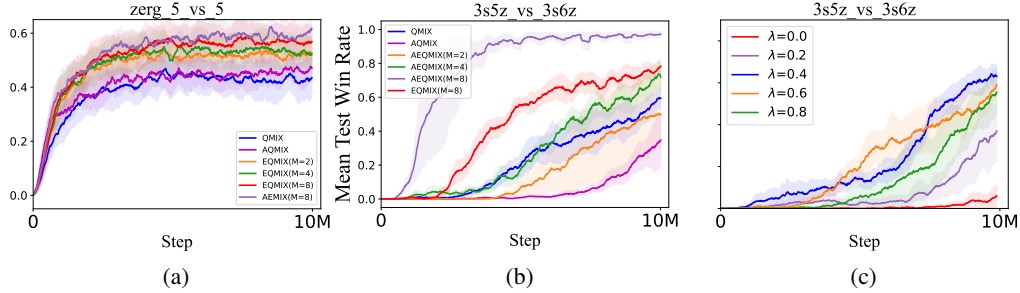

Figure 5: The performance on map (a) *zerg_5_vs_5* (b) *3s5z_vs_3s6z* with different anneal and ensemble combinations. (c) Mean test win rate of QMIX with different $\lambda$ on map *3s5z_vs_3s6z*.

## 6 Conclusion

This paper has brought attention to the often-overlooked issue of extrapolation error in MARL. We identified the critical role that value factorization methods play in mitigating this challenge and proposed additional techniques, such as multi-step bootstrapping and ensemble methods, to further reduce extrapolation error. Our experiments not only demonstrate the superior performance of our proposed approach but also validate the significant impact of extrapolation error on MARL performance. This success highlights the importance of addressing extrapolation error as a fundamental factor in improving MARL methods. We believe that our findings open up new avenues for advancing value factorization methods and offer a fresh perspective for future research in MARL.

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

## A    RELATED WORKS

**Value factorization**. In addition to the previously discussed methods, various approaches address the challenge of value factorization. For value-based methods, Qatten (Yang et al., 2020) employs an attention mechanism to augment the expressive capacity of linear factorization. WQMIX (Rashid et al., 2020a) improves QMIX's expressive ability through the incorporation of a weighted operator and a true value network. QTRAN++ (Son et al., 2020) refines the constraints of QTRAN to improve efficiency. For policy-based methods, VMIX (Su et al., 2021) applies QMIX's factorization to the value function of A2C (Mnih et al., 2016). DOP (Wang et al., 2020b) utilizes linear factorization on the Q-function of COMA (Foerster et al., 2018), while FOP (Zhang et al., 2021b) extend QPLEX's factorization to soft actor-critic (Haarnoja et al., 2018) framework. FACMAC (Peng et al., 2021) combines QMIX's factorization with MADDPG (Lowe et al., 2017). These approaches primarily concentrate on enhancing factorization itself, specifically addressing the TAE problem introduced in this paper, without delving into the underlying reasons for the success of value factorization. While theoretical papers such as (Wang et al., 2021) take steps to unveil the efficiency and credit assignment of value factorization, they lack substantial support for subsequent improvements. Recent examinations of these methods (Yu et al., 2022; Ellis et al., 2023; Hu et al., 2023), along with more comprehensive experiments, highlight QMIX as the most popular and robust value-based MARL algorithm. Therefore, distinct from previous approaches, our work approaches value factorization from a novel perspective, introducing further enhancements to existing methods.

**Ensemble RL and MARL**. Our analysis is similar to Averaged-DQN (Anschel et al., 2017), which ensembles the Q-functions from the past $M$ steps. Despite proving effective in variance reduction, Averaged-DQN relies on assumptions that may not always hold in practice. Other ensemble methods (Lee et al., 2021) incorporating the standard deviation of Q-functions were not discussed here due to limited observed improvements and their divergence from the main focus of this paper. REDQ (Chen et al., 2021) employs in-target minimization across a random subset of Q-functions from the ensemble. However, this approach proves unsuitable for value factorization, possibly due to the presence of model bias (See Appendix E.3). In MARL, EMAX (Schäfer et al., 2023) applied a similar ensemble method on VDN and QMIX with UCB and majority vote to improve exploration. MMD-MIX (Xu et al., 2021) introduce REM (Agarwal et al., 2020) into a distributional view of QMIX to be more robust in randomness. These methods do not explicitly consider the extrapolation error.

**Offline RL**. The in-target average ensemble employed in our paper bears resemblance to the approach used in offline RL (Agarwal et al., 2020; Fujimoto et al., 2019a; Levine et al., 2020). Additionally, the out-of-distribution (OOD) action studied in offline RL aligns with the extrapolation error addressed in this paper. As a result, similar methods may yield comparable effects. However, different from (An et al., 2021; Bai et al., 2022), our paper does not require a more conservative/pessimistic estimation of the target, as the monotonic constraint enables self-correction in online RL. Moreover, we found that any degree of pessimism negatively impacts performance, as detailed in the Appendix E.3.

## B    DETAILS IN SECTION 3

### B.1    EXTRAPOLATION ERROR IN MARL

In Section 3.1, we explore an illustrative example involving 5 agents, each with 10 possible actions. The scenario entails a uniform action selection for each agent at every state. The central question is: How many samples are required in each state to ensure the selection of all joint actions at least once? This scenario aligns with the classic coupon collector's problem in probability theory, where the expected number of samples needed grows asymptotically as $\mathcal{O}(n \log(n))$. The precise result is given by $n \sum_{i=1}^{n} \frac{1}{i}$. In the context of joint value functions, where $n = 10^5$, the requirement exceeds $10^6$ samples, whereas for factorized value functions with $n = 10$, only around 30 samples suffice.

Moving forward, we consider two similar examples: the maps *2s3z* and *5m_vs_6m* on the SMAC domain, both involving 5 agents with around 10 actions each. To calculate the proportion of unseen state-action pairs, for each transition $(s, \boldsymbol{a}, s')$, we mark $(s, \boldsymbol{a})$ as seen and then check whether $(s', \boldsymbol{a}')$ has been marked before, where $\boldsymbol{a}' = \arg\max Q(s', \boldsymbol{a}')$ is used for Q-learning update. The proportion is calculated from the $(s', \boldsymbol{a}')$ pairs that have not been seen in each batch of samples. This proportion

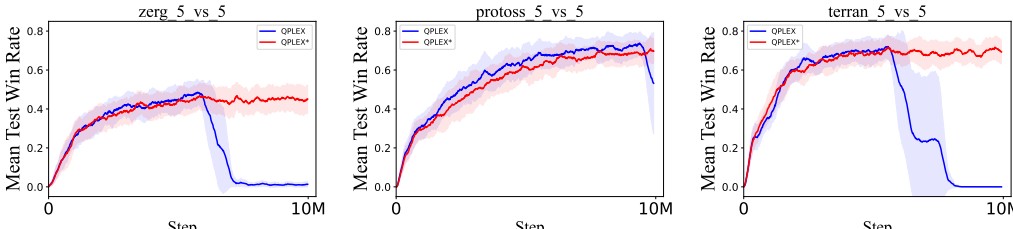

Figure 6: Comparison of the mean test win rate of QPLEX and QPLEX* on 3 maps of SMACv2. Plots show the mean and standard deviation across 3 seeds .

indicates how many $Q(s', a')$ values need to be extrapolated during each update. Note that the state space of SMAC is continuous and thus uncountable. We simplify it by directly employing $s = \text{int}(s)$. As a result, the actual proportion of unseen state-action pairs may be even higher than our calculated values.

To assess the TEE of QMIX in comparison to a centralized Q-function, we train the algorithm using samples generated from QMIX. Additionally, a centralized Q-function is trained, taking joint state and action as input and producing the corresponding Q-value as output. Given the computational challenges associated with obtaining a true Q-function on SMAC, we adopt an analytical approach akin to previous studies (Fujimoto et al., 2018; Chen et al., 2021). For each visited state-action pair, we compute the TEE by measuring the difference between the discounted Monte Carlo return and the estimated TD target. Since Monte Carlo return can change significantly throughout training, we normalize it by dividing the expected discounted return for state-action pairs sampled from the current policy.

### B.2 DETAILS OF QPLEX

Here, we present additional details of QPLEX as discussed in Section 3.3. Recall the factorization of QPLEX:

$$Q(s, \boldsymbol{a}) = \sum_i (\lambda_i(s, \boldsymbol{a}) - 1)(\hat{Q}_i(s, a_i) - \max_j \hat{Q}_j(s, a_j)) + \sum_i Q_i(s, a_i),$$

where $\hat{\ }$ indicates that the gradient is stopped. This factorization has two key properties: 1) $\frac{\partial Q}{\partial Q_i} = 1$, ensuring that the learning of individual utilities is unaffected by some certain poorly estimated $\lambda_i$. 2) $\max_a Q(s, \boldsymbol{a}) = \sum_i \max_{a_i} Q_i(s, a_i)$, which rules out the influence of $\lambda_i$ for optimal Q-functions.

However, because this factorization involves the joint action space, some joint actions $\boldsymbol{a}$ in $\lambda_i(s, \boldsymbol{a})$ may rarely be seen during training, leading to substantial errors. Moreover, since $\frac{\partial Q}{\partial \lambda_i} < 0$, under-estimation of the joint Q-function can lead to an overestimation of $\lambda_i$. As discussed in Section 3.2, underestimated values of $Q(s, \boldsymbol{a})$ make the corresponding joint action $\boldsymbol{a}$ less likely under state $s$, exacerbating the rarity of $(s, \boldsymbol{a})$ pairs. This leads to continuous overestimation of poorly estimated $\lambda_i(s, \boldsymbol{a})$, which can eventually cause instability in the learning process.

Our modification is $\lambda_i^*(s, \boldsymbol{a}) = \text{Sigmoid}(\lambda_i(s, \boldsymbol{a}))$, which prevents the instability from error accumulation. Additional results are presented in Fig. 6, showing that QPLEX* remains stable and performs similarly to the original QPLEX.

## C PROOF OF PROPOSITION 2

**Lemma C.1** ((Harutyunyan et al., 2016)). *The PQL operator can be rewritten in the following forms:*
$$\mathcal{N}_\lambda^{\mu,\pi} Q = (\mathcal{I} - \gamma\lambda\mathcal{P}^\mu)^{-1}(r + \gamma(1 - \lambda)\mathcal{P}^\pi Q). \tag{11}$$

Using this lemma, we have:
$$\mathcal{N}_\lambda^{\mu,\pi}(Q_k + e_k) = (\mathcal{I} - \gamma\lambda\mathcal{P}^\mu)^{-1}[r + \gamma(1 - \lambda)\mathcal{P}^\pi(Q_k + e_k)]$$
$$= \mathcal{N}_\lambda^{\mu,\pi} Q_k + \gamma(1 - \lambda)(\mathcal{I} - \gamma\lambda\mathcal{P}^\mu)^{-1}\mathcal{P}^\pi e_k.$$

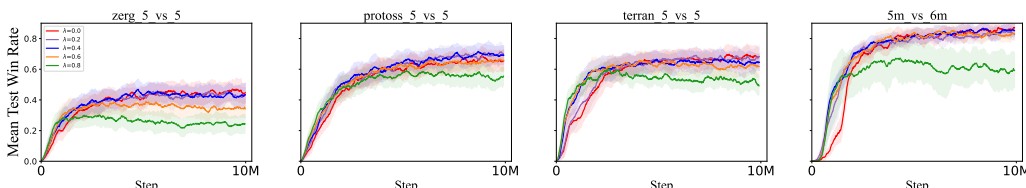

Figure 7: Mean test win rate of QMIX with different $\lambda$ on SMACv2 and SMAC.

As a result,

$$\|Q_{k+1} - \mathcal{N}_\lambda^{\mu,\pi}Q_k\|_\infty = \gamma(1-\lambda)\|(\mathcal{I} - \gamma\lambda\mathcal{P}^\mu)^{-1}\mathcal{P}^\pi e_k\|_\infty$$

$$\leq \frac{\gamma(1-\lambda)}{1-\gamma\lambda}\|e_k\|_\infty = \beta\epsilon.$$

This shows the propagation of TEE relative to $\lambda$ on each step.

For the algorithm:

$$\pi_k \in \boldsymbol{G}(Q_k) \ and \ Q_{k+1} = \mathcal{N}_\lambda^{\mu,\pi_k}Q_k + \varepsilon_k,$$

(Kozuno et al., 2021) introduced the following lemma:

**Lemma C.2** ((Kozuno et al., 2021))**.** *For any $K$ the following holds:*

$$\|V^{\rho\dagger} - V^{\rho_K}\| \leq \mathcal{O}(\beta^K) + \frac{2}{1-\gamma}\sum_{k=0}^{K-1}\beta^{K-k-1}\|\varepsilon_k\|_\infty \tag{12}$$

*where $\rho_K = \lambda\mu + (1-\lambda)\pi_k$, $\rho_\dagger = \lambda\mu + (1-\lambda)\pi_\dagger$ and $\pi_\dagger \in \boldsymbol{G}(Q^{\rho\dagger})$.*

Therefore, in this paper, we have

$$\|V^{\rho\dagger} - V^{\rho_K}\| \leq \mathcal{O}(\beta^K) + \frac{2}{1-\gamma}\sum_{k=0}^{K-1}\beta^{K-k-1}\cdot\beta\epsilon \tag{13}$$

The second term represents the error dependence which can be futher written as:

$$\frac{2}{1-\gamma}\sum_{k=0}^{K-1}\beta^{K-k}\epsilon = \frac{2\beta}{1-\gamma}\frac{1-\beta^K}{1-\beta}\epsilon = \mathcal{O}(\frac{\gamma(1-\lambda)}{(1-\gamma)^2}\epsilon).$$

This completes the proof.

## D  PSEUDO CODE

The pseudo code of AEQMIX is summarized in Algorithm 1.

## E  EXPERIMENTAL DETAILS

### E.1  IMPLEMENTATION

Our implementation of VDN, QMIX and QPLEX is based on the pymarl2 (Hu et al., 2023) code base. All hyper-parameters used in our algorithm is consistent with QMIX except for the additional $\lambda^*$ and ensemble size $M$, as presented in Table 2. Besides, the implementation of MADDPG and FACMAC is directly taken from Peng et al. (2021), without any parameter adjustment.

### E.2  FIGURES OF MAIN RESULTS

Fig. 8, 9 and 10 show the learning curves corresponding to the results presented in Table 1. In Fig. 10, we include the results of VDN, AEVDN, QPLEX* and AEQPLEX*, where (AE)QPLEX* corresponds to the variant proposed in Section 3.3. We can observe a substantial improvement in the AE-versions compared to the original algorithms.

---

**Algorithm 1** AEQMIX

---

1: Initialize $M$ action-value networks for all agents $\{[Q_i(\tau_i, a_i; \theta_i^j)]_{j=1}^M\}_{i=1}^n$ with parameter $\boldsymbol{\theta}$ and a mixing hypernetwork $H$ with parameter $\psi$
2: Initialize target networks: $\psi' = \psi$, $\boldsymbol{\theta}' = \boldsymbol{\theta}$
3: Initialize replay buffer $\mathcal{D} = \{\}$
4: **while** $k \leq episode\_max$ **do**
5:    set trajectory buffer $T = [\,]$
6:    **for** $t = 1$ to $max\_epsode\_length$ **do**
7:       Explore using $\varepsilon - greedy$ with $\bar{Q}_i(\tau_i, \cdot) = \sum_{j=1}^M Q_i(\tau_i, \cdot; \theta_i^j)$
8:       Store transition $(s_t, \boldsymbol{\tau}_t, \boldsymbol{a}_t, r_t, s_{t+1}, \boldsymbol{\tau}_{t+1})$ into $T$
9:    **end for**
10:   Store trajectory into $\mathcal{D}$ and sample a mini-batch $b$
11:   **for** each trajectory $T$ in $b$ **do**
12:      **for** each transition $(s, \boldsymbol{\tau}, \boldsymbol{a}, r, s', \boldsymbol{\tau}')$ in $T$ **do**
13:         Form joint action $\boldsymbol{a}'$ by $a_i' = \arg\max \bar{Q}_i(\tau_i, \cdot)$
14:         Compute target joint value $\bar{Q}'(s', \boldsymbol{a}')$ using (9)
15:      **end for**
16:      Compute PQL target $y_{s,a}$ with $\lambda$ using $\bar{Q}'(s', \boldsymbol{a}')$
17:      Compute joint value $Q(s, \boldsymbol{a}; \boldsymbol{\theta}^j, \psi)$
18:   **end for**
19:   Compute loss through (10)
20:   Adam updates $\boldsymbol{\theta}$, $\psi$ with the computed loss
21:   **if** $k \bmod d = 0$ **then**
22:      Update target networks: $\psi' = \psi$, $\boldsymbol{\theta}' = \boldsymbol{\theta}$
23:      Update $\lambda$ through (8)
24:   **end if**
25:   $k = k + 1$
26: **end while**

---

Table 2: Hyperparameters used for SMAC, SMACv2 and GRF.

| hyperparameters | SMAC | SMACv2 | GRF |
|---|---|---|---|
| Action Selector | epsilon greedy | epsilon greedy | epsilon greedy |
| $\epsilon$ start | 1.0 | 1.0 | 1.0 |
| $\epsilon$ finish | 0.05 | 0.05 | 0.05 |
| $\epsilon$ Anneal Time | 100000 | 100000 | 100000 |
| Runner | parallel | parallel | parallel |
| Batch Size Run | 8 | 4 | 32 |
| Buffer Size | 5000 | 5000 | 2000 |
| Batch Size | 128 | 128 | 128 |
| Optimizer | Adam | Adam | Adam |
| Target Update Interval | 200 | 200 | 200 |
| Mixing Embed Dimension | 32 | 32 | 32 |
| Hypernet Embed Dimension | 64 | 64 | 64 |
| Learning Rate | 0.001 | 0.001 | 0.0005 |
| $\lambda$ | 0.6 | 0.4 | 0.8 |
| $\lambda^*$ | {0.0, 0.4} | {0.0, 0.2} | 0.8 |
| Ensemble Size | 8 | 8 | 2 |
| Gamma | 0.99 | 0.99 | 0.999 |
| RNN Hidden Dim | 64 | 64 | 256 |

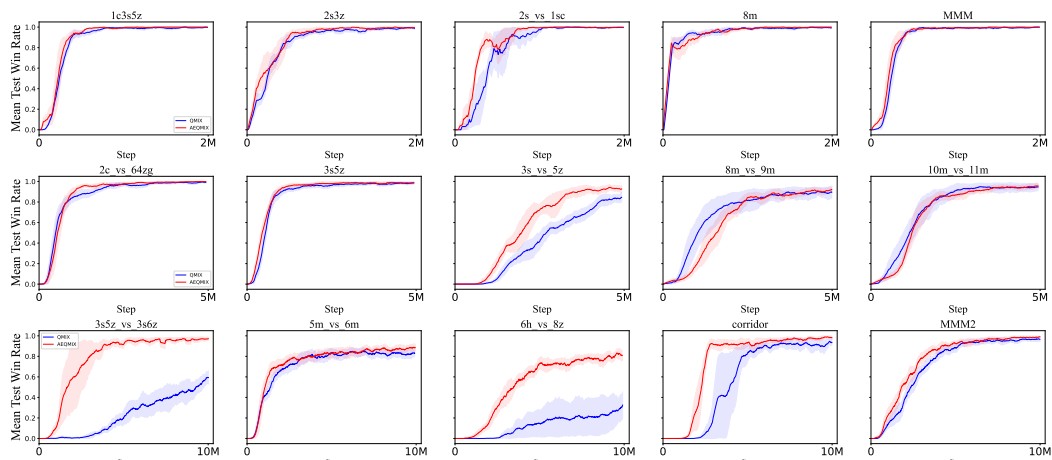

Figure 8: Comparison of the mean test win rate of QMIX and AEQMIX on SMAC. Plots show the mean and std across 3 seeds .

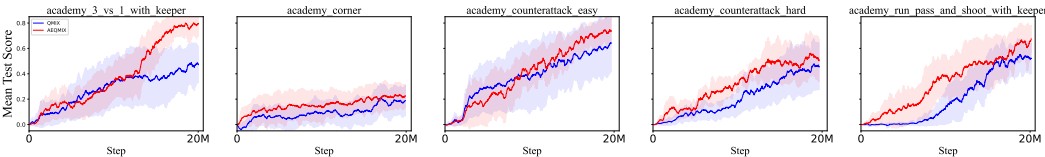

Figure 9: Comparison of the mean test score of QMIX and AEQMIX on GRF. Plots show the mean and std across 5 seeds .

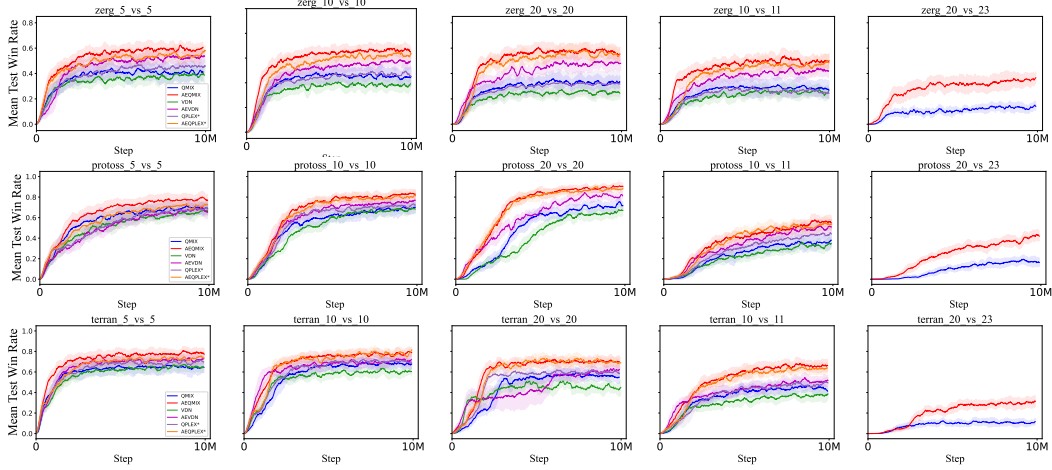

Figure 10: Comparison of the mean test win rate of QMIX, AEQMIX, VDN, AEVDN, QPLEX* and AEQPLEX* on SMACv2. Plots show the mean and std across 3 seeds.

### E.3    ADDITIONAL RESULTS ON ENSEMBLED TARGET

First, we test the effect of using multiple mixing networks to compute the ensembled target:

$$Q(s, \boldsymbol{a}; \boldsymbol{\theta}, \bar{\psi}) = \sum_{j=1}^{M} H_j(s, Q_1(s, a_1; \theta_1), ..., Q_n(s, a_n; \theta_n); \psi^j). \tag{14}$$

As shown in Fig. 11, using $M > 1$ mixing networks does not benefit the performance. This is because the extrapolation error mainly arises from the action space and the mixing network only takes the state as input.

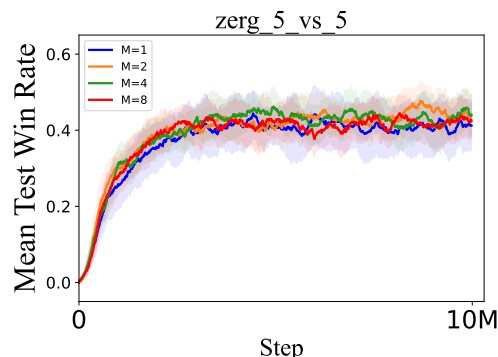

Figure 11: Comparison of the performance with different ensemble size on the mixing network of QMIX.

Then, we test using REDQ's (Chen et al., 2021) target:

$$y = r + \gamma \min_{j \in \mathcal{M}} Q(s, \boldsymbol{a}; \boldsymbol{\theta}^j, \psi) \tag{15}$$

where $\mathcal{M}$ is a set of $M$ distinct indices from the ensemble $\{1, 2, ..., 10\}$. The result is shown in Fig. 12, where the degree of pessimism decreases as the x-axis increases. We can observe that either pessimistic or optimistic will reduce the performance. This is probably because the model bias in QMIX will also be underestimate or overestimate through the additional $\min$ or $\max$ operator.

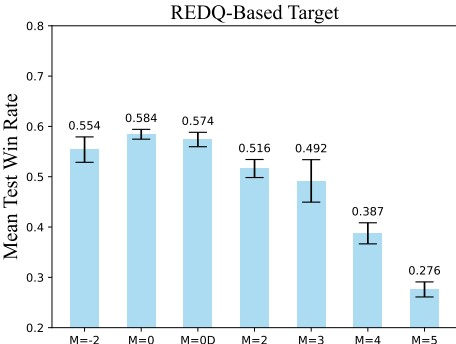

Figure 12: Comparison of the performance with different REDQ-based target. Plots show the mean and standard deviation averaged from zerg_5_vs_5, protoss_5_vs_5 and terran_5_vs_5 on SMACv2 at 3M time steps. $M = 0$ represents average over all indices. $M = 0D$ represents average while using double q-learning to sample actions. $M = -2$ represents use $\max$ instead of $\min$ in (15).

### E.4 ADDITIONAL RESULTS OF FACMAC

Fig. 13 shows the additional results regarding AEFACMAC and FACMAC on SMAC.

### E.5 ADDITIONAL RESULTS ON GRF

Fig. 14 shows the importance of using a large $\lambda$ on performance of GRF.

In Section 4.2, the choice is made to mix the individual utilities before averaging them. We liken this approach similar to the target policy smoothing in TD3. As shown in Fig. 15, variants of our method are compared, including the original "mix first" approach used in AEQMIX, an "average first" one and an "average first with dropout" one. In the latter, dropout is incorporated into the mixing network.

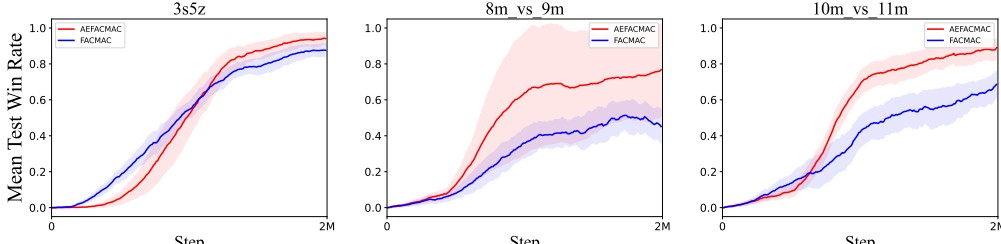

Figure 13: Comparison of the mean test win rate of AEFACMAC and FACMAC on SMAC.

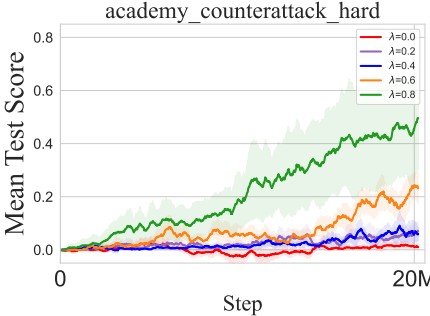

Figure 14: Comparison of the mean test score of QMIX with different $\lambda$ on GRF.

The comparison reveals that both the *mix first* and *average first + dropout* approaches outperform the *average first* approach alone. This observation leads to the conclusion that introducing noise into the input has a similar effect to dropout in terms of regularizing the mixing network.

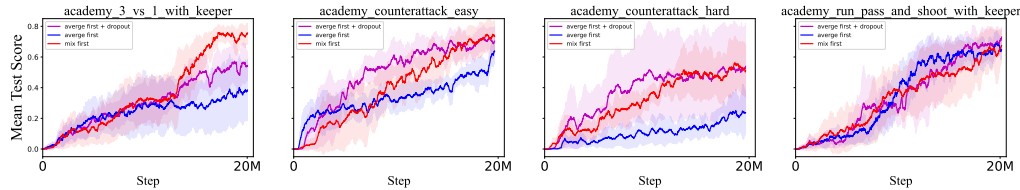

Figure 15: Comparison of the mean test score of several variants of AEQMIX on GRF.

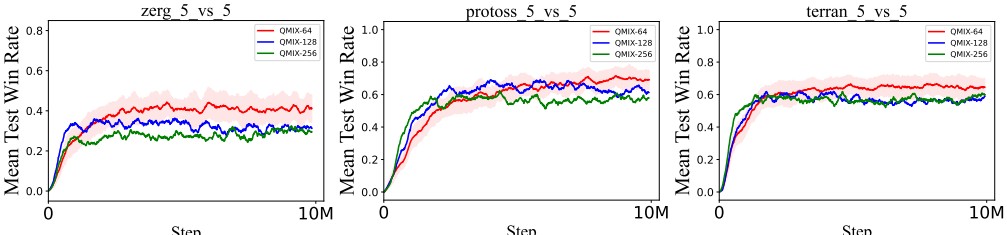

Figure 16: Comparison of the mean test win rate of QMIX with different hidden size.

