# OpenReview forum: "Addressing Extrapolation Error in Multi-Agent Reinforcement Learning"
_ICLR.cc/2025/Conference — Submitted to ICLR 2025_

### Official Review · Reviewer_h49b · 2024-10-28

**Soundness:** 2
**Presentation:** 1
**Contribution:** 2
**Rating:** 3
**Confidence:** 4

**Summary:**

This paper discusses extrapolation error in multi-agent reinforcement learning (MARL). The authors show that extrapolation error is a critical issue in MARL, affecting performance due to propagation from unseen state-action pairs, especially when the action space is large, as is often the case in MARL. Instead of proposing a new algorithm, the authors introduce two existing techniques, annealed multi-step bootstrapping and ensembled TD targets, to mitigate extrapolation error. The proposed method is tested across three domains: SMAC, GRF and SMACv2. The results show that the two simple modifications to existing methods can lead to significant performance improvements.

**Strengths:**

- Extrapolation error in MARL is a natural extension of the single-agent case to the multi-agent case, which is reasonable.
- The improved method is tested across three domains on numerous maps, which is commendable.

**Weaknesses:**

- Lack of novelty. Although the paper does not introduce new techniques or methods, I would not consider this a lack of novelty. The lack of novelty in this paper lies in its discussion of extrapolation error, which does not offer anything new. Extrapolation error is a commonly discussed topic in single-agent RL and naturally extends to MARL, which is acceptable. However, the authors do not provide new insights or discussions about challenges specific to MARL. Most of the content is similar to the single-agent case. It feels more like stitching together existing works [1,2,3,4] rather than proposing a new perspective or addressing a new issue induced by the multi-agent setting.

- The writing needs improvement. The core idea is simple and natural, but the logic is messy. There are many statements that are too subjective without any evidence, making the paper less convincing. For example, line 352 states, "The behavior policy typically originates from the old policy stored in the replay buffer, which may not align closely with the current policy after convergence". Does this not hold even after convergence? Why? Additionally, some conclusions are not consistent with the results. For example, line 294 states, "While the mean and standard deviation of $\lambda$ remain small, the maximum value of $\lambda$ grows significantly as training progresses, eventually leading to performance degradation." However, Fig 2 shows that $\lambda_{max}$ decreases over time, and the performance increases when $\lambda_{max}$ increases.

- Too many results are placed in the appendix but are referenced in the main text, especially since some claims are based on the appendix (e.g., lines 355, 407, 411, and 430). This affects the readability of the paper.

- The paper claims that extrapolation error is a major issue in MARL, but the authors do not provide any evidence to support this claim. The two proposed techniques are for bias/variance reduction, which do not seem to be directly related to extrapolation error. There is no evidence that the proposed method mitigates extrapolation error, thus leading to better performance.

- There are no implementation details or parameter searches provided for the baseline methods. Only searching parameters for the proposed method is unfair and may lead to biased results.

- Minor issue. Some learning curves are missing in the last column of Appendix Figure 10.

[1] Fujimoto, Scott, David Meger, and Doina Precup. "Off-policy deep reinforcement learning without exploration." In International conference on machine learning, pp. 2052-2062. PMLR, 2019.


[2] Anschel, Oron, Nir Baram, and Nahum Shimkin. "Averaged-dqn: Variance reduction and stabilization for deep reinforcement learning." In International conference on machine learning, pp. 176-185. PMLR, 2017.


[3] Rashid, Tabish, Mikayel Samvelyan, Christian Schroeder De Witt, Gregory Farquhar, Jakob Foerster, and Shimon Whiteson. "Monotonic value function factorisation for deep multi-agent reinforcement learning." Journal of Machine Learning Research 21, no. 178 (2020): 1-51.


[4] Kozuno, Tadashi, Yunhao Tang, Mark Rowland, Rémi Munos, Steven Kapturowski, Will Dabney, Michal Valko, and David Abel. "Revisiting Peng’s Q ($\lambda$) for Modern Reinforcement Learning." In International Conference on Machine Learning, pp. 5794-5804. PMLR, 2021.

**Questions:**

- Since the paper discusses extrapolation error in MARL, can you provide results demonstrating that your method mitigates extrapolation error compared to the baselines?

- Since the Target Estimation Error (TEE) can be influenced by issues such as overestimation and extrapolation errors, how can you ensure that the issue is indeed extrapolation error due to unseen state-action values backpropagating rather than overestimation due to the max operator [1]?

- Section 3 provides a detailed analysis of QPLEX to illustrate extrapolation error in MARL. However, Section 4 switches to QMIX. Is there a specific reason for this switch?

- See Weaknesses.

[1] Anschel, Oron, Nir Baram, and Nahum Shimkin. "Averaged-dqn: Variance reduction and stabilization for deep reinforcement learning." In International conference on machine learning, pp. 176-185. PMLR, 2017.

---

> ### Author Response · Authors · 2024-11-20
>
> Thanks for your review. We answer this question first, which helps with the subsequent rebuttal
> > **How can we ensure that extrapolation error is reduced?**
>
> We approach this from both theoretical and empirical perspectives:
>
> **Theoretical Perspective**: Extrapolation error arises when the target Q-function relies on values from rarely seen actions. By reducing the usage of such values, we can mitigate extrapolation error. For example: Factorized Q-functions operate in a much smaller action space compared to joint Q-functions. With the same sample size, factorized Q-functions are less likely to use extrapolated Q-values, reducing the extrapolation error. This theoretical property highlights why value factorization methods effectively mitigate extrapolation error.
>
> **Empirical Perspective**:
> While extrapolation error **cannot be directly measured** due to its integration within the total neural network error, we use Target Estimation Error (TEE), as introduced in our paper, as an indirect indicator.
> For example, since value factorization theoretically reduces extrapolation error, its impact on TEE provides indirect evidence.
> As shown in figure 1(b)(c), TEE is significantly reduced. Although value factorization introduces limitations in function approximation, which should increase TEE, the observed reduction in TEE confirms that the theoretical and empirical results mutually validate each other.
>
> > **1. Extrapolation error is a commonly discussed topic in single-agent RL and naturally extends to MARL.**
>
> We’d like to emphasize that it is not straightforward to consider extrapolation error in **online MARL**. Unlike offline single/multi-agent RL, where extrapolation error has been studied extensively, it is rarely considered in online single-agent RL and has not been addressed in online MARL prior to our work. The unique challenge in MARL is its large joint action space, which amplifies extrapolation error. This is discussed in Section 3.1 of our paper.
>
> > **2. The authors do not provide new insights or discuss challenges specific to MARL.**
>
> We respectfully argue that our paper introduces several **new insights** and all **specific to MARL**:
> - Online MARL suffers from exacerbated extrapolation error due to large joint action spaces.
> - Value factorization methods effectively mitigate extrapolation error by addressing joint action space challenges.
> - Monotonic factorization is crucial to prevent the accumulation of extrapolation error.
> - Performance in existing MARL methods is heavily influenced by extrapolation error.
>
> None of these points can be directly derived from prior works [1,2,3,4], demonstrating that our findings are novel and specific to MARL.
>
> > **3. The paper claims that extrapolation error is a major issue in MARL, but the authors do not provide any evidence to support this claim.**
>
> Here, we summarize the evidence provided based on the 4 points mentioned above, respectively.
> - In Figure 1(a), we conduct experiments on SMAC and find 20%-60% of the target estimation relies on extrapolated values, directly highlighting the importance of extrapolation error in MARL.
> - As discussed at the beginning, value factorization reduces extrapolation error by limiting the use of extrapolated Q-values. This theoretical property is empirically supported in Figures 1(b)(c), where TEE decreases significantly.
> - Section 3.2 provides theoretical analysis showing that monotonic factorization is critical to prevent extrapolation error accumulation. Although experiments specific to this are not included, prior works [5,6] on non-monotonic factorization support our claim.
> - In Section 3.3, we show how extrapolation error destabilizes QPLEX. Furthermore, Section 5 and the appendix demonstrate that methods such as VDN, QMIX, QPLEX, FACMAC, and MADDPG show substantial performance improvement when extrapolation error is addressed.
>
> > **4. Line 352 states, "The behavior policy typically originates from the old policy stored in the replay buffer, which may not align closely with the current policy after convergence". Does this not hold even after convergence? Why?**
>
> Sorry for this mistake. We meant “before convergence” rather than "after convergence."

---

> > ### Author Response · Authors · 2024-11-20
> >
> > >**5. Line 294 states, "While the mean and standard deviation of $\lambda$ remain small, the maximum value of $\lambda$ grows significantly as training progresses, eventually leading to performance degradation."**
> >
> > As shown in Figure 2, $\lambda_\max$ significantly increase from 0 to 15 during the first 5M steps while $\lambda_{mean}$ and $\lambda\_{std}$ remain below 0.5. This trend indicates that large $\lambda$ values only appear for a small subset of joint actions. This behavior aligns with extrapolation error: errors accumulate on these rarely updated joint actions, ultimately causing instability in QPLEX’s training at around 5M steps.
> >
> > To confirm that the observed growth of $\lambda_\max$ is problematic, we introduced QPLEX*, which directly bounds $\lambda$ to the range [0,1]. QPLEX* achieves the same performance without the instability observed in standard QPLEX. This result validates that the unbounded growth of $\lambda$ is the root cause of the performance degradation.
> > The subsequent decline of $\lambda_\max$ after 5M steps is irrelevant, as the training has already crashed at that point.
> >
> > >**6. Too many results are placed in the appendix but are referenced in the main text, especially since some claims are based on the appendix  (e.g., lines 355, 407, 411, and 430).**
> >
> > We acknowledge that referencing results in the appendix can hinder readability. For the next version, we will:
> > - Relocate key results from the appendix (e.g., those referenced in lines 355 and 430) to the main text.
> > - Clarify that implementation details (e.g., those referenced in lines 407 and 411), while important, are secondary to the main results and thus remain in the appendix.
> >
> > > **7. We address the following concerns collectively:**
> > >- The two proposed techniques are for bias/variance reduction, which do not seem to be directly related to extrapolation error.
> > >- There is no evidence that the proposed method mitigates extrapolation error, thus leading to better performance.
> > >- Since the paper discusses extrapolation error in MARL, can you provide results demonstrating that your method mitigates extrapolation error compared to the baselines?
> > >- Since the Target Estimation Error (TEE) can be influenced by issues such as overestimation and extrapolation errors, how can you ensure that the issue is indeed extrapolation error due to unseen state-action values backpropagating rather than overestimation due to the max operator [1]?
> >
> > **Do the proposed methods reduce extrapolation error?**
> > Both techniques—PQL and ensemble methods—directly address extrapolation error:
> > - PQL: Reduces extrapolation error by assigning lower weights to target Q-values that may rely on extrapolated values.
> > - Ensemble: By lowering the variance of the target Q-function, ensemble methods reduce extrapolation error as part of the total error reduction.
> >
> > This leads us to investigate whether the performance improvements stem from **extrapolation error reduction** or **other errors** addressed by the proposed methods. Since extrapolation error cannot be directly measured, we rely on TEE, which mainly captures both extrapolation and overestimation errors. Figure 4 shows that TEE is significantly reduced with our methods. To validate that this reduction and performance improvements stems from extrapolation error:
> > - **Ruling out Overestimation**: Appendix E3 (Figure 12) shows that introducing a more conservative target (to reduce overestimation) negatively impacts performance. This demonstrates that overestimation is not a major factor for the tested methods, confirming that the observed performance gain arises from mitigating extrapolation error.
> > - **Testing Mixing Networks**: Ensemble methods applied to the mixing network reduce general error, but they do not improve performance (Figure 11). This is because extrapolation error arises from the action space, which the mixing network does not influence.
> > - **Structural Bias vs. Extrapolation Error**: While PQL may also reduce structural bias in value factorization methods, its effectiveness is not limited to such cases. Both QPLEX and MADDPG—methods without structural bias—benefit significantly from our techniques. This supports the conclusion that the observed performance gains primarily result from reduced extrapolation error.
> >
> > We used complementary approaches to isolate the impact of extrapolation error on performance. The results, supported by both **theoretical analysis** and **empirical validation**, demonstrate that our methods effectively reduce extrapolation error and significantly enhance performance. We believe that there are no other significant issues that may affect performance.

---

> ### Author Response · Authors · 2024-11-20
>
> > **8. There are no implementation details or parameter searches provided for the baseline methods. Only searching parameters for the proposed method is unfair and may lead to biased results.**
>
> The baselines are implemented using fine-tuned versions from PyMARL2 (VDN, QMIX, QPLEX) and FACMAC’s paper (FACMAC, MADDPG). We did not perform parameter searches for any methods, including ours, as $\lambda^*$ and $M$ were set heuristically.
>
> > **9. Section 3 provides a detailed analysis of QPLEX to illustrate extrapolation error in MARL. However, Section 4 switches to QMIX. Is there a specific reason for this switch?**
>
> We chose QPLEX for theoretical analysis due to its explicit modeling of joint action spaces. QMIX was used in Section 4 because of its simplicity and popularity. However, the findings in Section 4 are applicable to other methods, including QPLEX.
>
> [1] Fujimoto, Scott, David Meger, and Doina Precup. "Off-policy deep reinforcement learning without exploration." In International conference on machine learning, pp. 2052-2062. PMLR, 2019.
>
> [2] Anschel, Oron, Nir Baram, and Nahum Shimkin. "Averaged-dqn: Variance reduction and stabilization for deep reinforcement learning." In International conference on machine learning, pp. 176-185. PMLR, 2017.
>
> [3] Rashid, Tabish, Mikayel Samvelyan, Christian Schroeder De Witt, Gregory Farquhar, Jakob Foerster, and Shimon Whiteson. "Monotonic value function factorisation for deep multi-agent reinforcement learning." Journal of Machine Learning Research 21, no. 178 (2020): 1-51.
>
> [4] Kozuno, Tadashi, Yunhao Tang, Mark Rowland, Rémi Munos, Steven Kapturowski, Will Dabney, Michal Valko, and David Abel. "Revisiting Peng’s Q ($\lambda$) for Modern Reinforcement Learning." In International Conference on Machine Learning, pp. 5794-5804. PMLR, 2021.
>
> [5] Hu, Jian, et al. "Rethinking the Implementation Tricks and Monotonicity Constraint in Cooperative Multi-agent Reinforcement Learning." The Second Blogpost Track at ICLR 2023.
>
> [6] Peng, Bei, et al. "Facmac: Factored multi-agent centralised policy gradients." Advances in Neural Information Processing Systems 34 (2021): 12208-12221.

---

> > ### Comment · Reviewer_h49b · 2024-11-25
> >
> > Thank you for your response. However, my concerns remain unaddressed. I will keep my score unchanged.
> >
> > 1. Novelty. Although the authors argue that their paper introduces several new insights, I still find these insights to be obvious for me and are from the literature I mentioned.
> >
> > 2. Evidence of extrapolation error. It is still not convincing that extrapolation error is the main reason for the performance gap. The error definition in eq.(3) is the same as the one in [1], except for the name change from "overestimation error" to "TEE". I do not see evidence in the analysis that extrapolation error is the main reason for the performance gap.
> >
> > 3. Experiments. The authors mentioned that they did not tune the hyperparameters for baselines and the proposed method. This is not a good practice, and the sensitivity of the proposed method to hyperparameters is unclear. I recommend tuning the hyperparameters for all methods and reporting the results in the next version, at least for common hyperparameters like learning rate and the specific hyperparameters for the proposed method.
> >
> > [1] Anschel, Oron, Nir Baram, and Nahum Shimkin. "Averaged-dqn: Variance reduction and stabilization for deep reinforcement learning." In International conference on machine learning, pp. 176-185. PMLR, 2017.

---

### Official Review · Reviewer_JVoU · 2024-10-31

**Soundness:** 3
**Presentation:** 3
**Contribution:** 3
**Rating:** 6
**Confidence:** 3

**Summary:**

The authors discuss and provide an analysis on the extrapolation error in Multi-Agent Reinforcement Learning (MARL), and show that value factorisation methods, like QMIX, can help reduce this error. Furthermore, they propose two methods to reduce extrapolation error in MARL, specifically multi-step bootstrapping and using ensembled independent value functions. The authors show that these methods can improve the performance of QMIX, in SMAC, SMACv2 and Google Research Football (GRF) environments, and of on-policy MARL algorithms like MADDPG and FACMAC on SMAC.

**Strengths:**

- The paper is well-written, clear and easy to follow.
- Extrapolation error, especially in online MARL, is a relatively unexplored area. This paper appears to be among the first to address this issue, providing both an analysis and methods to mitigate it.
- The paper provides a relevant discussion on the propagation of extrapolation error in MARL and how value factorization methods can help reduce it. Building on this analysis, the authors introduce targeted modifications to reduce the bias and variance associated with extrapolation error, with results showing consistent performance improvements across different environments and algorithms. Additionally, ablation studies on ensemble size and the $\lambda$ annealing parameter are included.

**Weaknesses:**

- The experiment section is not very detailed. The authors should provide more information on the experimental setup, including how many seeds were run, the evaluation procedure and the evaluation interval. Furthermore, the main results presented in Table 1 don't include the standard deviation, which is important to understand the significance of the results.
- Although the authors provide results in three environments, two of them are SMAC and SMACv2, which might share some similarities. It might be more informative to use a different environment to SMACv1.
- It is unclear if parameter sharing is used in the baseline algorithms. If it is, then the proposed ensemble method would result in many more learnable parameters. This could be a potential source of the improvement in the results, especially since when using smaller ensembles ($M=1,2$) in Figure 5b, performance is worse than the QMIX baseline. It would be important to disentangle the effect of increased capacity and extrapolation error mitigation.

**Questions:**

1. Was parameter sharing used in the baselines?
2. What is the comparison of the parameter counts across the baseline and proposed modifications? How does this scale as the number of agents increase?
3. Could the authors specify the experiment details as discussed in the weaknesses.

---

> ### Author Response · Authors · 2024-11-20
>
> We sincerely thank the reviewer for their detailed and constructive feedback. We are also grateful for the recognition of the strengths of our work, including the clarity of the paper, our novel focus on extrapolation error in online MARL, and the thorough analysis and methods we introduced to mitigate this issue.
>
> > **1. The experiment section is not very detailed.**
>
> We appreciate the reviewer’s concern about the level of detail in the experimental section. The number of seeds is detailed in Figures 8, 9, and 10 in the appendix, where SMAC and SMACv2 use 3 seeds, and GRF uses 5 seeds.
>
> The evaluation procedure follows the standard setup for SMAC: training pauses every $10^4$ steps, and the agent is evaluated for 32 episodes using greedy action selection.
>
> Regarding the omission of standard deviations in Table 1, this decision was made to save space. However, the learning curves in Figures 8, 9, and 10 in the appendix provide a detailed view of performance, including error bar, which we believe addresses this concern.
>
> > **2. It might be more informative to use a different environment to SMACv1.**
>
> Thank you for the suggestion. We are currently conducting experiments on several tasks from PettingZoo to expand the diversity of environments. Unfortunately, due to the time required for tuning and running baselines, we are unable to provide these results during the rebuttal phase. We appreciate your understanding.
>
> > **3. It is unclear if parameter sharing is used in the baseline algorithms.**
>
> We confirm that all baseline algorithms (except for MADDPG) utilize parameter sharing across agents in our experiments.
>
> > **4. Why using smaller ensembles (M=1,2) in Figure 5b, performance is worse than the QMIX baseline.**
>
> It is due to the annealing of $\lambda$ occurring too quickly, before convergence is achieved, and not because of the ensemble itself.
> As discussed in Lines 510–515, the annealing approach is designed for quicker convergence. However, as shown in Figure 5(c), smaller $\lambda$ leads to poor performance on the 3s5z_vs_3s6z task due to insufficient convergence. Premature annealing of $\lambda$ negatively impacts performance in these cases. In contrast, larger ensembles achieve convergence earlier, making $\lambda$ -annealing more effective.
> Additionally, Figure 5(a) demonstrates that smaller ensembles (M=2) still significantly improve performance compared to the baseline.
>
> > **5. It would be important to disentangle the effect of increased capacity and extrapolation error mitigation.**
>
> Thank you for this valuable suggestion. To address this concern, we conducted experiments with larger QMIX models on SMACv2 by increasing the hidden dimensions of the individual Q-function from 64 to 128, and 256. The results (Figure 16) indicate that simply increasing model capacity does not improve performance, confirming that the observed improvements are not due to increased capacity.
>
> > **6. What is the comparison of the parameter counts across the baseline and proposed modifications? How does this scale as the number of agents increase?**
>
> The primary increase in parameter count arises from the ensemble modification.
>
> - In QMIX, the total parameter count is _individual_Q_parameters + mixer_parameters_. With an ensemble of size M, this scales to _M*individual_Q_parameters + mixer_parameters_.
> - For policy-based methods like FACMAC, the baseline parameters include _individual_Q_parameters + individual_policy_parameters + mixer_parameters_, which scale to _M×individual_Q_parameters + individual_policy_parameters + mixer_parameters_ with an ensemble of size M.
>
> The parameter count depends on the size of each component and the state-action space of the task. For example, in the _corridor_ task of SMAC, QMIX’s individual Q-parameters total 39k, while the mixer parameters are 69k. Importantly, the parameter count does not scale with the number of agents due to parameter sharing, except in methods like QPLEX and MADDPG, which take joint actions as input.

---

> > ### Comment · Reviewer_JVoU · 2024-11-25
> >
> > Thank you to the authors for their responses and clarifications. While the rebuttal addresses some concerns and strengthens the paper, I will maintain my initial score and still recommend a weak accept.

---

### Official Review · Reviewer_nWE9 · 2024-11-03

**Soundness:** 2
**Presentation:** 2
**Contribution:** 2
**Rating:** 3
**Confidence:** 3

**Summary:**

This paper addresses the challenge of extrapolation errors in multi-agent reinforcement learning (MARL), focusing on the issue caused by the large joint action space. To mitigate these issues, the authors propose the application of modified multi-step bootstrapping and ensemble TD target techniques, aiming to enhance learning stability and reduce prediction variance. These proposed solutions are supported by theoretical analysis that explains the propagation of extrapolation errors and the importance of ensuring consistent value estimation. Empirical results validate these approaches, demonstrating that they contribute to improved performance and more stable training in various MARL scenarios.

**Strengths:**

- Insightful Theoretical Analysis: The theoretical framework helps illustrate the propagation of extrapolation errors and lays a foundation for understanding the importance of stable value estimation in MARL.
- The proposed methods, including multi-step bootstrapping and ensemble TD targets, are backed by experiments showing improved performance and stability over baseline approaches in MARL settings, demonstrating their utility in practice.
- The paper highlights the extrapolation errors in MARL and proposes practical solutions to mitigate this challenge, contributing to a better understanding and partial resolution of this important problem.

**Weaknesses:**

- Fundamental Limitations of Value Factorization: Although the paper claims that the success of value factorization methods is largely due to their ability to mitigate extrapolation errors (as noted in the abstract), this mitigation is not comprehensive. The approach simplifies the estimation by focusing on local utilities, but it may fail to capture the full complexity of joint action spaces. An agent’s action value can vary significantly when combined with other agents’ actions, leading to potential suboptimal solutions. While this method improves learning stability, as further discussed in Sections 3.1 and 3.2, it does not fully address the diverse combinations and dependencies between agents, which are critical for optimal policy learning in MARL.
- Incremental and Limited MARL-Specific Solutions: The proposed methods, while addressing the large joint action space, primarily adapt existing techniques like multi-step bootstrapping and ensemble TD targets. These approaches lack innovation and do not sufficiently consider agent interactions, a key aspect of MARL. This results in simplified solutions that may fall short in effectively handling complex, cooperative scenarios, limiting their overall impact and applicability.

**Questions:**

While the paper notes that value factorization mitigates extrapolation errors, how does the method address the potential suboptimality caused by not fully capturing the complexity of joint action interactions among agents? Are there plans to extend the method to better account for agent dependencies and interaction effects?

---

> ### Author Response · Authors · 2024-11-20
>
> We thank the reviewer for their thoughtful feedback. We greatly appreciate the recognition of our paper’s insightful theoretical analysis, the practicality of our proposed methods, and the contribution to mitigating extrapolation errors in MARL. These strengths highlight the significance of our work in providing a deeper understanding of stable value estimation and its role in improving MARL performance and stability.
>
> > **1. Fundamental Limitations of Value Factorization.**
>
> We acknowledge the fundamental limitations of value factorization in its reliance on a factorized action space. However, addressing this issue comprehensively has remained a challenge for years and is **not the primary focus of our paper**. Despite its potential suboptimality, value factorization consistently demonstrates superior performance compared to other baselines. Our objective is to investigate **why it performs well and identify ways to further exploit this advantage**.
>
> Our findings provide insights that:
>
> - **Support Value Factorization in addressing joint action dependencies**: We show that ignoring extrapolation errors during attempts to capture the full joint action space can result in significant error accumulation. For example, our modified QPLEX* improves upon QPLEX by considering joint action spaces while accounting for extrapolation errors. This enhances stability without compromising performance.
>
> - **Generalize beyond Value Factorization**: The insights derived from our study can be directly applied to methods like MADDPG, which do not suffer from suboptimality issues, or inspire improvements in these methods.
>
> > **2. Incremental and Limited MARL-Specific Solutions.**
>
> We emphasize that the primary goal of our paper is to build a _theoretical foundation_ for understanding and mitigating extrapolation errors in MARL, rather than introducing purely novel methods. To this end, we deliberately use **well-established techniques** with **clear theoretical properties** to validate our insights. We believe that _addressing a critical, overlooked issue using simple and effective approaches represents a meaningful contribution to the field_.
>
> While we acknowledge that our methods do not explicitly consider agent interactions, we do offer **MARL-specific considerations**:
> - We introduced QPLEX*, which bounds QPLEX's $\lambda(s,a)$ to avoid extrapolation error accumulation and improves the stability.
> - We find that starting with a large $\lambda$ in PQL, while being unusual in single-agent settings due to suboptimality, actually benefits MARL due to the extrapolation error.
> - In line 406-408, we opt not to ensemble the mixing network since it is unrelated to extrapolation, as verified in Figure 11.
> - In line 408-412, we avoid averaging before the mixing network to implicitly regularize it, as verified in Figure 15.
>
> > **3. Addressing Suboptimality and Agent Dependencies**
>
> As noted above, our method does not directly address the suboptimality inherent in value factorization caused by its inability to fully capture joint action interactions. However, our contributions can:
> - **Facilitate future developments in value factorization methods**: Previous attempts to address suboptimality, such as QTRAN and QPLEX, struggled with extrapolation errors, which we mitigate through our approach. Our insights can guide the development of methods that aim to overcome suboptimality while retaining stability.
> - **Benefit MARL methods without suboptimality issues**: Since our study addresses extrapolation errors—a common challenge across MARL methods—it can directly apply to or inspire enhancements in approaches that do not rely on value factorization.
>
> We hope these responses clarify our contributions and demonstrate how our work can drive progress in MARL research.

---

> > ### Comment · Reviewer_nWE9 · 2024-11-24
> >
> > Thank you for your detailed and thoughtful response. I appreciate the effort you put into addressing my concerns and clarifying the contributions of your work. However, after considering your explanation, I still believe that the contributions are not sufficient to warrant acceptance. I'd like to maintain my score.

---

### Meta-Review · Area_Chair_sGFo · 2024-12-14

**Metareview:**

This paper is about reducing extrapolation error in team-reward cooperative MARL setting as captured by SMAC and similar environments. Extrapolation error is defined as arising from states not encountered during training receiving unrealistic values after learning. Reviewers felt that the work was incremental, and not novel enough, especially that multi-agent aspects of the setting (interactions between agents) were not really addressed in a way that pushed things forward beyond what was already done in the large literature which now exists on cooperative MARL.

**Additional Comments On Reviewer Discussion:**

All reviewers responded to say that they were not convinced by the authors' replies to them.

The authors sent me a message to complain about the reviewers not changing their scores, but I don't find it convincing. The reviewers' comments look much more plausible to me than the strange things the authors claim in their message: that this paper with MARL in the title really wasn't meant to be about MARL? I doubt it.

---

### Decision · Program_Chairs · 2025-01-22

Reject